# OpenReview forum: "Classification Attention for Chinese NER"
_ICLR.cc/2020/Conference — Reject_

### Official Review · AnonReviewer2 · 2019-10-22
**Official Blind Review #2**

**Rating:** 3

**Review:**

Summary:
      This paper discussed an approach to do named entity resolution (NER, the paper focuses only on Chinese NER but I think it could generalize to other languages as well). The idea is based on smart integration and extension of multiple existing building blocks: 1) BERT pre-trained model 2) a previous work to get document embedding by doing weighted average of word embedding (https://openreview.net/pdf?id=SyK00v5xx) and 3) Scaled dot-product attention mechanism applied directly to multi-label classification. The "Introduction", "Related work", and "Experiment Settings" sections are well written and covers many details and decent references. Especially, the "experiments" section is described in a great amount of details, which should be very helpful for reproducibility.

Contributions:
      * The author found an interesting application of the original algorithm (https://openreview.net/pdf?id=SyK00v5xx) to represent the entity class embedding based on averaging "BERT" embeddings of all the component words. This could be implemented as a pre-processing step against any training dataset to derive "pre-learned" entity class embedding.
      * Instead of the common approach of connecting the BERT sequence outputs directly to CRF layer, the author added an intermediate layer to calculate the classification attention between a sentence (sequence of token embedding) and any entity class (based on the above pre-learned entity embedding). This result plus the original sentence embedding are concatenated.  The concatenation is further fed into a few additional layers to produce the final inputs into CRF layer.

Weakness:
     * The paper lacks novelty. As pointed above, I did not see that the contribution from the paper is sufficiently original. It is a good application of various existing methods though.

I also have a few suggestions/questions below:

* The ERNIE paper (https://arxiv.org/abs/1907.12412v1) is mentioned in the related work. Since ERNIE can potentially learn a good vocab for Chinese, did you ever compare your approach vs ERNIE+CRF?
* There is one paper that I know which is pretty relevant to what you are doing here, which is probably worth a reference. https://arxiv.org/abs/1805.04174.  In that paper, the idea is to co-learn a class embedding and perform text classification. Their class attention is performed through dot-production attention though.
* The Table index seems wrong in your paper. (I think Table 2 is not mentioned in your paper, but all tables (3-6) is offset by 1).
* There are some minor typos or places that need some clarifications.
   - in the abstract: "character-based" model. This is a little confusing. Because BERT is a word-piece based model. word-piece could across multiple characters for English. IIUC, You probably want to say "Chinese-character" instead of character.
   - in "Introduction", "providing greater weight to characters identical to each entity class", you might want to revise this sentence to clarify its meaning further.
   - In section 3.2, you might want to give some explanation to some notations (the first time you refer to it). For example, what is $L$, what is $m$ and $n$.  What is $S$?  Also why the denominator of Emb(Word) is not $n-m+1$?

   - The last paragraph in section 3.3 needs more clarification as well. How do you merge the three tensors after attention stage? (a concatenation ?) . The last sentence mentioned "residential", I guess instead you want to say "residual".  You might also want to clarify where the "3 layers" of residual appear in your network.

  - In your experiment, (if I did not miss), did you freeze the BERT parameters and entity embeddings when finetuning your NER model?

  - in Table 2 and Table 3, why the 13-layer BERT + CRF performs significantly worse on Recall (Table 2) and significantly better on Recall (Table 3)?




**Experience Assessment:**

I have read many papers in this area.

**Review Assessment: Checking Correctness Of Derivations And Theory:**

I carefully checked the derivations and theory.

**Review Assessment: Checking Correctness Of Experiments:**

I carefully checked the experiments.

**Review Assessment: Thoroughness In Paper Reading:**

I read the paper thoroughly.

---

> ### Public Comment · ~Ge_Yuchen1 · 2019-11-08
> **Some Explanation about your confusion.**
>
> thank you so much for your kind review. Here gives answsers to all of your questions
>
> 1、	Result compare with ERNIE+CRF
>         Baidu did not publish the pre-training model for the Ernine 2.0, but from the paper, the results of MSRA are as follows:
>        Ernie 1.0 Base    f1:93.8
>        Ernie  2.0 Base   f1:93.8
>        Ernie  2.0 large  f1:95.0
>        Also,we are trying to conduct experiment of Ernine 1.0 +CRF
>
> 2、	Similarities with published papers 《Joint Embedding of Words and Labels for Text Classification》
>        Thanks to the reviewer's recommendation, but I had not read any papers about label embeddings in NLP including the paper《Joint Embedding of Words and Labels for Text Classification》before.
>        The idea that giving greater weight to characters in the statement that are similar to a particular entity type was gradually formed in working with NER task. The original source of inspiration came from the Lattice paper https://arxiv.org/abs/1805.02023. I found it pretty useful to combine word embeddings to the character-level model. However, the effect of the whole model has a lot to do with the completeness of predefined lexicon (word level) and pretrained word embedding. So, I tried to eliminate the step of obtaining predefined lexicon, and directly replace the external resources with the easy-to-obtain entity class embeddings.
>      After reading this article many times in the past two days, I found that although the results of both methods are weighted representations of certain words in the article, the process of achieving this result is two completely different approaches.
>      In the paper《Joint Embedding of Words and Labels for Text Classification》, the label embeddings are trained together with the model, hoping to obtain a class template that could highlight related words with the particular text Topic .For instance, in Figure4(a),it highlights “coaches”、”sports” for STORT news and “rock”, “drummer” for Entertainment news.
>       However, our goal is to highlight word closed to predefined names-entity classes only. The class embeddings are fixed at the initial stage of training
>       For instance, suppose there is a series of Olympic news and entertainment news that needs to be classified. According to the model proposed by《Joint Embedding of Words and Labels for Text Classification》，words like “gymnasium”、”swimming pool”、”gold medal” may also be highlighted along with PERSON- related words “athlete”, “coaches”. However, if the predefined entity class in our method is only PERSON, entities in the remaining categories will not be highlighted under ideal conditions
>       In my opinion, our approach has a more dominant effect on the NER field, or in areas with a defined reading goal, because the class embedding for attention is preset to the entity class that you want to extract. But label embedding also gives potential entity information from the opposite perspective. In the future, we can even try to combine the two methods and discover more potential entities, especially for Internet data!
>      As can be seen from the experimental department, compared to the regular datasets, our method works especially well for user generated data like Weibo, which also shows the validity of class attention in searching potential entities.
>     Finally, I would like to thank the reviewers for recommending the paper《Joint Embedding of Words and Labels for Text Classification》. We will continue to follow other articles in this direction.
>
> 4,Sorry for the problems of Table index and notations. They will be fixed in the next version.
>
> 5,We did not freeze the pretrained BERT model, but we freeze all the class embeddings.
>
> 6,- in Table 2 and Table 3, why the 13-layer BERT + CRF performs significantly worse on Recall (Table 2) and significantly better on Recall (Table 3)?
> This is the result of a real experiment. My own guess is the irregularity of Weibo data.

---

### Official Review · AnonReviewer3 · 2019-11-04
**Official Blind Review #3**

**Rating:** 3

**Review:**

This paper tries to improve the performance of Chinese NER by developing a novel attention mechanism that leverages BERT pre-trained model which considers bi-directional context. Experiments on a number of tasks show that the proposed approach is effective.

Comments:
[1] A bunch of experiments are conducted
[2] Chinese NER is a hard problem, but it would be great to see the proposed approach generalizable to other tasks. So, the contribution of this paper is limited
[3] The proposed algorithm is simple and effective, but the novelty is a bit low


**Experience Assessment:**

I do not know much about this area.

**Review Assessment: Checking Correctness Of Derivations And Theory:**

I did not assess the derivations or theory.

**Review Assessment: Checking Correctness Of Experiments:**

I assessed the sensibility of the experiments.

**Review Assessment: Thoroughness In Paper Reading:**

I read the paper at least twice and used my best judgement in assessing the paper.

---

> ### Public Comment · ~Ge_Yuchen1 · 2019-11-07
> **Thank you for your comment**
>
> Thank you for your kind suggestion,
> we realized that we should clearly state our potential value in the next version
> I think the Classification Attention structure may have other uses in the future, although this article is merely focusing on the contribution to the NER task only.
>
> For example, in the field of textual understanding (Text classification, QA and so on), explicit entities displayed in the sentence can bring reading focus to the text, and we will try to apply this structure in other fields in future work.
>
> For innovation
> Our approach proposes two innovations
> 1.	It proposed a simple but effective way to obtain entity class embeddings from the pre-trained BERT model. This method avoids the need for additional lexicon (avoiding OOV problem) and has strong practicability.
> 2.	It proposed a structure to use word level information (entity class embeddings) to locate protentional characters in the sentence that may compose an entity
> 	It can be applied to other language with similar language structure with Chinese. （Chinese word can be composed of one or a couple of characters and that word boundaries cannot detected graphically）
> 	And this structure is different from the scale dot product in terms of both purpose and structure. It can display the position of the potential entity characters in the highlighted statement, which is useful for subsequent tasks.
> 	from the experiment section, especially the user-generated network data (i.e Weibo dataset), we have made a big improvement to the traditional method, which proves the effectiveness of using class embedding to find potential entities in the statement.

---

### Official Review · AnonReviewer1 · 2019-11-04
**Official Blind Review #1**

**Rating:** 3

**Review:**

Comments by sections :

Summary :

The use of entity is not clear. Are you referring to named-entities ?

1 INTRODUCTION

"Due to the differences in language structure" : this paragraph is not clear. It should say explicitly that in Chinese word can be composed of one or a couple of characters and that word boundaries can not detected graphically.

 "A common mitigation is to add external lexicon as a reference"   references are needed on how to includes words in embedding (except just training word embeddings)

 " an entity classification-assisted model" : not very clear : the classification is assisted ?

 " The first step is to form word embeddings of entities appearing in the trainning sentences through character embeddings and the second step is to aggregate entity embeddings by category and generate classification embeddings" : is it a multi-task training ?

 "After that, we designed a novel Attention mechanism to integrate entity " : is it the same model or two different propositions of the paper ? "After that" is not very clear as a transition.

 Section 2 :

 "more works Yang et al. (2016) Ruder12 et al. (2017) " : strange formulation

 "What’s more, the attention mechanism..." : odd expression.

 "In this paper, we revise the Scaled Dot-Product Attention to Classification Attention which would give a weighted representation of the input sentences through a series of entity classes." : maybe it should be moved to the introduction as a novelty proposed by the paper.


 3.2  EMBEDDING EXTRACTION FOR ENTITY CLASS

 " the smooth inverse frequency is abandoned" : why ? please explain this choice.
 "the weighted projection of the word embeddings on their first singular vector is removed." : explain. If it is a common practice, give a citation otherwise justify this choice.

 3.3 CLASSIFICATION ATTENTION

 Here again, the proposed attention system is described but not justified : why would a class specific attention system be better ? What are the expected advantages ?

 4.1.3 EXPERIMENTAL RESULTS
 Experiments are conducted on 4 dataset and the proposed model is compared to a "standard" BERT-based model and several results form the litterature. The proposed model outperforms sometimes the other models, often by a small margin as it is usually the case in NER experiments.
 But more insight on the strengths of the models should be given by conducting an ablation study.


 In conclusion ,this paper present an incremental improvement over BERT-based NER for Chinese. The proposed approach is not sufficiently justified and experiments, even if showing improvements over state-of-the-art models or published results, does not sufficiently explore the benefits of the proposed model (with ablation study for example).

**Experience Assessment:**

I have read many papers in this area.

**Review Assessment: Checking Correctness Of Derivations And Theory:**

N/A

**Review Assessment: Checking Correctness Of Experiments:**

I assessed the sensibility of the experiments.

**Review Assessment: Thoroughness In Paper Reading:**

I read the paper at least twice and used my best judgement in assessing the paper.

---

> ### Public Comment · ~Ge_Yuchen1 · 2019-11-07
> **Some Explanation about your confusion.**
>
> First of all, thank you for your kind review. Before answering your questions, I think some description in my article is indeed misleading for non-Chinese readers. The next version will be carefully modified. Here is a common explanation for you:
>
> A Chinese word is usually composed of multiple Chinese characters, and there is no interval between Chinese word or Chinese characters in a sentence, so it is very difficult to detect the boundary for the NER problem.
>
> A simple Example can be given like:
>
> English：I   heart   Beijing
> Chinese：我爱北京
> Chinese CWS: 我（I）/爱(heart) /北京(Beijing)
>
> Just like Beijing refers to 北京 in Chinese，the word 北京 is composed of 2 Chinese characters 北 and 京. In English, There are obvious boundaries between words , but in Chinese, all the Chinese characters are written in a whole.
>
> The following is an explanation of some confusing words in the article.
>
> 1、”character” in this paper should refer to “Chinese character”,  （eg. 北）
> 2、“word” in this paper should refer to “Chinese word “.（eg. 北京）
> 2、	“word entity” in this paper should refer to “named-entity “in the paper
> 3、“Entity class embedding“ in this paper refer to a single representative embedding of all the named-entities within a certain class （eg. embedding for class People）
> 4、“lexicon“ refer to “user defined dictionary（in Chinese word level）”
>
> #Summery
> entity is referring to named-entities
>
> #1 Introduction
> "Due to the differences in language structure" : this paragraph is not clear. It should say explicitly that in Chinese word can be composed of one or a couple of characters and that word boundaries can not detected graphically.
> Yes, I should say "in Chinese word can be composed of one or a couple of characters and that word boundaries cannot detected graphically" directivity
>
>
> "A common mitigation is to add external lexicon as a reference"
> The next sentence behind it “Whether directly integrated into character-based models by means of word embeddings, or directly repairing the boundaries of entities recognized by character-based models, the performance of lexicon-assisted model is strongly related to the quality of the lexicon.” gives the idea of how to include Chinese word embedding into Chinese character model
>
> " an entity classification-assisted model”
> Perhaps I should say the assistance from entity classes.
>
> " The first step is to form word embeddings of entities appearing in the training sentences through character embeddings and the second step is to aggregate entity embeddings by category and generate classification embeddings"
> As illustrated in 3.1 OVERALL ARCHITECTURE," the embedding extraction step is performed firstly in a pipeline structure and the results will be aggregated to the attention model."
>
> "After that, we designed a novel Attention mechanism to integrate entity "
> We have already explained that we have proposed two proposals, the last paragraph is explaining the first proposal(which consists of several steps), and this paragraph is the second, and the two proposals are formed by the pipeline structure.
>
> Section2
> more works Yang et al. (2016) , Ruder12 et al. (2017) refers to two paper
> Sorry for the missing comma.
>
> "What’s more, the attention mechanism..."
> Each paragraph in this section gives a way to promote the traditional model, and this paragraph gives the attention mechanism used in Chinese NER
>
> 3.2 EMBEDDING EXTRACTION FOR ENTITY CLASS
> " the smooth inverse frequency is abandoned"
> The reason is in the next sentence” due to the dynamic semantic information encoded in H. “
>
> "the weighted projection of the word embeddings on their first singular vector is removed." : explain. If it is a common practice, give a citation otherwise justify this choice.
> The reference to this method has already been mentioned in the previous article. “Inspired from Arora et al. (2016), in this paper, we illustrate a simple way to extract word embeddings of entities in the hidden outputs of BERT.”
>
> 3.3
> Here again, the proposed attention system is described but not justified : why would a class specific attention system be better ? What are the expected advantages ?
>
> Because this attention mechanism can use the extracted class embedding to focus on the potential entities of the same type in the text. It is explained in the paper: It aims to get a series of weighted representation of the input sentences so that characters similar to a particular entity class could have greater feature
>
> 4.ablation study.
> We designed two baseline experiments to do the ablation study, one is pure bert+crf to eliminate the superiority of the bert , and the other is a 13-layer bert (the original bert is 12) Layer) +crf, used to eliminate the extra layer of scaled dot product attention itself, showing that the structure of our model : the original bert of 12 layer + 1 layer of our proposed class attention + crf is working

---

### Decision · Program_Chairs · 2019-12-19

**Decision:**

Reject

**Comment:**

The paper is interested in Chinese Name Entity Recognition, building on a BERT pre-trained model. All reviewers agree that the contribution has limited novelty. Motivation leading to the chosen architecture is also missing. In addition, the writing of the paper should be improved.